# Phylogenetic variation in cortical layer II immature neuron reservoir of mammals

Chiara La Rosa[1,2], Francesca Cavallo[1], Alessandra Pecora[1], Matteo Chincarini[3], Ugo Ala[2], Chris G Faulkes[4], Juan Nacher[5], Bruno Cozzi[6], Chet C Sherwood[7], Irmgard Amrein[8,9], Luca Bonfanti[1,2]*

[1]Neuroscience Institute Cavalieri Ottolenghi (NICO), Orbassano, Italy; [2]Department of Veterinary Sciences, University of Turin, Torino, Italy; [3]Università degli Studi di Teramo, Facoltà di Medicina Veterinaria, Teramo, Italy; [4]School of Biological and Chemical Sciences, Queen Mary University of London, London, United Kingdom; [5]Neurobiology Unit, BIOTECMED, Universitat de València, and Spanish Network for Mental Health Research CIBERSAM, València, Spain; [6]Department of Comparative Biomedicine and Food Science, University of Padova, Legnaro, Italy; [7]Department of Anthropology and Center for the Advanced Study of Human Paleobiology, The George Washington University, Washington DC, United States; [8]D-HEST, ETH, Zurich, Switzerland; [9]Institute of Anatomy, University of Zurich, Zurich, Switzerland

*For correspondence:
luca.bonfanti@unito.it

Competing interests: The authors declare that no competing interests exist.

**Abstract** The adult mammalian brain is mainly composed of mature neurons. A limited amount of stem cell-driven neurogenesis persists in postnatal life and is reduced in large-brained species. Another source of immature neurons in adult brains is cortical layer II. These cortical immature neurons (cINs) retain developmentally undifferentiated states in adulthood, though they are generated before birth. Here, the occurrence, distribution and cellular features of cINs were systematically studied in 12 diverse mammalian species spanning from small-lissencephalic to large-gyrencephalic brains. In spite of well-preserved morphological and molecular features, the distribution of cINs was highly heterogeneous, particularly in neocortex. While virtually absent in rodents, they are present in the entire neocortex of many other species and their linear density in cortical layer II generally increased with brain size. These findings suggest an evolutionary developmental mechanism for plasticity that varies among mammalian species, granting a reservoir of young cells for the cerebral cortex.

## Introduction

Structural changes occurring in the adult brain are important for physiological plasticity (adaptation to changing environment), protection against age-related dysfunction (e.g., dementia), and possibly brain repair (*Martino et al., 2011*; *Bond et al., 2015*; *Bao and Song, 2018*). Brain structural plasticity consists of synaptic plasticity (synapse formation/elimination; *Forrest et al., 2018*) and genesis of new neurons driven by neural stem cells (*Aimone et al., 2014*; *Lim and Alvarez-Buylla, 2016*). The latter process, known as adult neurogenesis, is widely present in the brain of non-mammalian vertebrates (including most of the telencephalon; *Ganz and Brand, 2016*) and becomes spatially restricted to a few, subcortical neurogenic niches in mammals (*Bond et al., 2015*; *Lim and Alvarez-Buylla, 2016*; *Aimone et al., 2014*; *Feliciano et al., 2015*). Adult neurogenesis is substantially absent in the neocortex, with only a very small amount of postnatal addition of interneurons described in mice (*Dayer et al., 2005*). The highly expanded neocortex of large-brained mammals is located far from the neurogenic sites. Additionally, the mammalian neocortex requires substantial

**eLife digest** To acquire new skills or recover after injuries, the mammalian brain relies on plasticity, the ability for the brain to change its architecture and its connections during the lifetime of an animal.

Creating new nerve cells is one way to achieve plasticity, but this process is rarer in humans than it is in mammals with smaller brains. In particular, it is absent in the human cortex: this region is enlarged in species with large brains, where it carries out complex tasks such as learning and memory. Producing new cells in the cortex would threaten the stability of the structures that retain long-term memories.

Another route to plasticity is to reshape the connections between existing, mature nerve cells. This process takes place in the human brain during childhood and adolescence, as some connections are strengthened and others pruned away.

An alternative mechanism relies on keeping some nerve cells in an immature, 'adolescent' state. When needed, these nerve cells emerge from their state of arrested development and 'grow up', connecting with the appropriate brain circuits. This mechanism does not involve producing new nerve cells, and so it would be suitable to maintain plasticity in the cortex. Consistent with this idea, in mice some dormant nerve cells are present in a small, primitive part of the cortex.

La Rosa et al. therefore wanted to determine if the location and number of immature cells in the cortex differed between mammals, and if so, whether these differences depended on brain size. The study spanned 12 mammal species, from small-brained species like mice to larger-brained animals including sheep and non-human primates.

Microscopy imaging was used to identify immature nerve cells in brain samples, which revealed that the cortex in larger-brained species contained more adolescent cells than its mouse counterpart. The difference was greatest in a region called the neocortex, which has evolved most recently. This area is most pronounced in primates – especially humans – where it carries out high-level cognitive tasks.

These results identify immature nerve cells as a potential mechanism for plasticity in the cortex. La Rosa et al. hope that the work will inspire searches for similar reservoirs of young cells in humans, which could perhaps lead to new treatments for brain disorders like dementia.

structural stability, which is thought to be related to the retention of long-term memories (*Parolisi et al., 2018*; *Koketsu et al., 2003*).

Recently, attention has been focused on a different population of cortical cells that might also be involved in plasticity, the 'immature' neurons (cINs; *Gómez-Climent et al., 2008*; *Piumatti et al., 2018*; *Rotheneichner et al., 2018*; *Benedetti et al., 2020*), which are generated prenatally, but continue to express typical markers of immaturity during adulthood, including doublecortin (DCX; *Gómez-Climent et al., 2008*; *Luzzati et al., 2009*) and 'polysialylated' or 'embryonic' Neural Cell Adhesion Molecule (PSA-NCAM; *Seki and Arai, 1991*). The cINs can progressively mature through the lifespan, ultimately losing the markers for immaturity (*Rotheneichner et al., 2018*; *Benedetti et al., 2020*). They are considered as a potential reservoir of young, plastic neuronal phenotypes (*Piumatti et al., 2018*; *Bonfanti and Nacher, 2012*; *La Rosa et al., 2019*), which might represent a form of slow, delayed neurogenesis ('neurogenesis without division') if ultimately integrated into circuits (*Rotheneichner et al., 2018*; *Benedetti et al., 2020*; *König et al., 2016*). These immature cells were initially discovered in cortical layer II of the mouse and rat piriform cortex (reviewed in *Bonfanti and Nacher, 2012*). While in rodents they are confined to the paleocortex (three-layered allocortex), some mammals also host them in the neocortex (six-/five-layered isocortex; *Luzzati et al., 2009*; *Zhang et al., 2009*; *Cai et al., 2009*). Current knowledge on this cell population is fragmentary, due to the complexity of systematic studies involving large-sized and wild animal species (*Bonfanti and Nacher, 2012*; *La Rosa et al., 2019*). The recent qualitative observation that cINs are widespread in the cerebral cortex of sheep, a mammal endowed with a relatively large and gyrencephalic brain (*Piumatti et al., 2018*), invites the hypothesis that they might be important more generally in mammalian species with larger brain sizes than rodents' (*Piumatti et al., 2018*; *Palazzo et al., 2018*). To test this idea, here we systematically studied the occurrence, anatomical

distribution, morphology, protein expression profile, maturity/immaturity state, and density (linear density: number of DCX+ cells/mm of cortical layer II) of cINs in the cerebral cortex of 80 brains across a phylogenetic diversity of eutherian mammals (spanning three of the four mammalian super-orders; *Nishihara et al., 2009*) which are widely different in their neuroanatomy (brain size, gyrencephaly, encephalization; *Figure 1—figure supplement 1* and *Supplementary file 2*) and other life history and socioecological features (lifespan, habitat, food habit).

## Results

To analyse cINs in a systematic and comparable way across mammals with different neuroanatomies and brain sizes, we identified homologous brain structures to define four correspondent, anterior-posterior levels in each species (L1-L4; see *Figure 1D* and Methods). Structural plasticity is an age-dependent process, usually decreasing over the lifespan (*Ben Abdallah et al., 2010*; *Varea et al., 2009*). However, it is challenging to identify perfectly equivalent maturational states across a phylogenetically diverse sample of species (*Snyder, 2019*; *Workman et al., 2013*). There is conflicting information about the lifespan of different species (considering average lifespan, maximum longevity and relative age are various options; *Justice et al., 2016*), and it is difficult to precisely match a specific age in one species to a 'corresponding' age in another species (*Snyder, 2019*; *Workman et al., 2013*). We used the average lifespan (data available in Animal Diversity Web; *Myers et al., 2019*, University of Michigan); the species-typical lifespans have been split into prepuberal and adult stages, the latter further divided into three stages: young-adult, middle age and aged (*Figure 1C*), by adapting the classification of the American Veterinary Medical Association for six life stages in dogs (*Bartges et al., 2012*). The main goal of our study was to investigate cINs in adults; prepuberal stages were only considered to explore the age-related decline around puberty, as previously described for adult neurogenesis (*Ben Abdallah et al., 2010*).

### Characterization of DCX+ cortical neurons across mammals

Though postmortem interval (PMI; the time between death and fixation of the brain), fixation procedure and other conditions can influence the detection of DCX+ cells in brain tissues, as described for neurogenesis in the hippocampus (*Chawana et al., 2020*; *Kempermann et al., 2018*), no substantial variation was observed in our specimens in the quality and intensity of DCX staining of cINs (*Figures 2* and *3*), regardless of slight differences in the fixation procedures across specimens in the sample (*Supplementary file 1*). The PMI length, considered as one of the main limits to brain tissue quality (*Moreno-Jiménez et al., 2019*), was generally very short (between a few minutes and 1 hr in all specimens analysed here, apart from chimpanzee, which was less than 14 hr; *Supplementary file 1*). In order to confirm that we were localizing the cINs as they have been described previously in rodents and sheep (*Gómez-Climent et al., 2008*; *Rubio et al., 2016*; *Piumatti et al., 2018*; reviewed in *Bonfanti and Nacher, 2012*; *König et al., 2016*) in all species studied, unbiased by differences in fixation procedure, we considered multiple features of this neuronal population, such as: i) the morphology of the DCX+ cells (type 1 and type 2 cells, based on cell soma size and dendritic arborisation); ii) the staining of other markers (PSA-NCAM, NeuN); iii) the possible co-expression with markers for cell proliferation. In addition, we checked for the occurrence of staining for all markers in other anatomical regions of the same animals where DCX expression is known to be at a high level: the SVZ and hippocampal neurogenic zones (*Figure 3A*), and the piriform cortex (*Figure 2—figure supplement 1*).

The anatomical distribution of the DCX+ neurons was investigated in both paleo- and neocortex (*Figure 2B*). DCX expression was consistently found in cells located in the upper part of layer II of the paleocortex, in all the species considered, at all ages evaluated (*Figure 2—figure supplement 1*). In rodents (mouse and naked mole rat), only occasional DCX+ neurons were detectable in layer II of the neocortex. In other small-brained, lissencephalic species (sengi and bats) they were also found in the lateral part of the neocortex, with variable distribution along the brain dorsal-ventral axis. In all other mammals in our study, DCX+ neurons were observed throughout the entire neocortex (*Figure 2B*). Spatial distributions were rather similar in the different anterior-posterior brain levels (*Figure 4—figure supplement 1A*).

The morphology of the layer II DCX+ cells fell into the two main types in all mammals investigated, as reported previously (type 1, small cell soma, bipolar; type 2, large cell soma with ramified

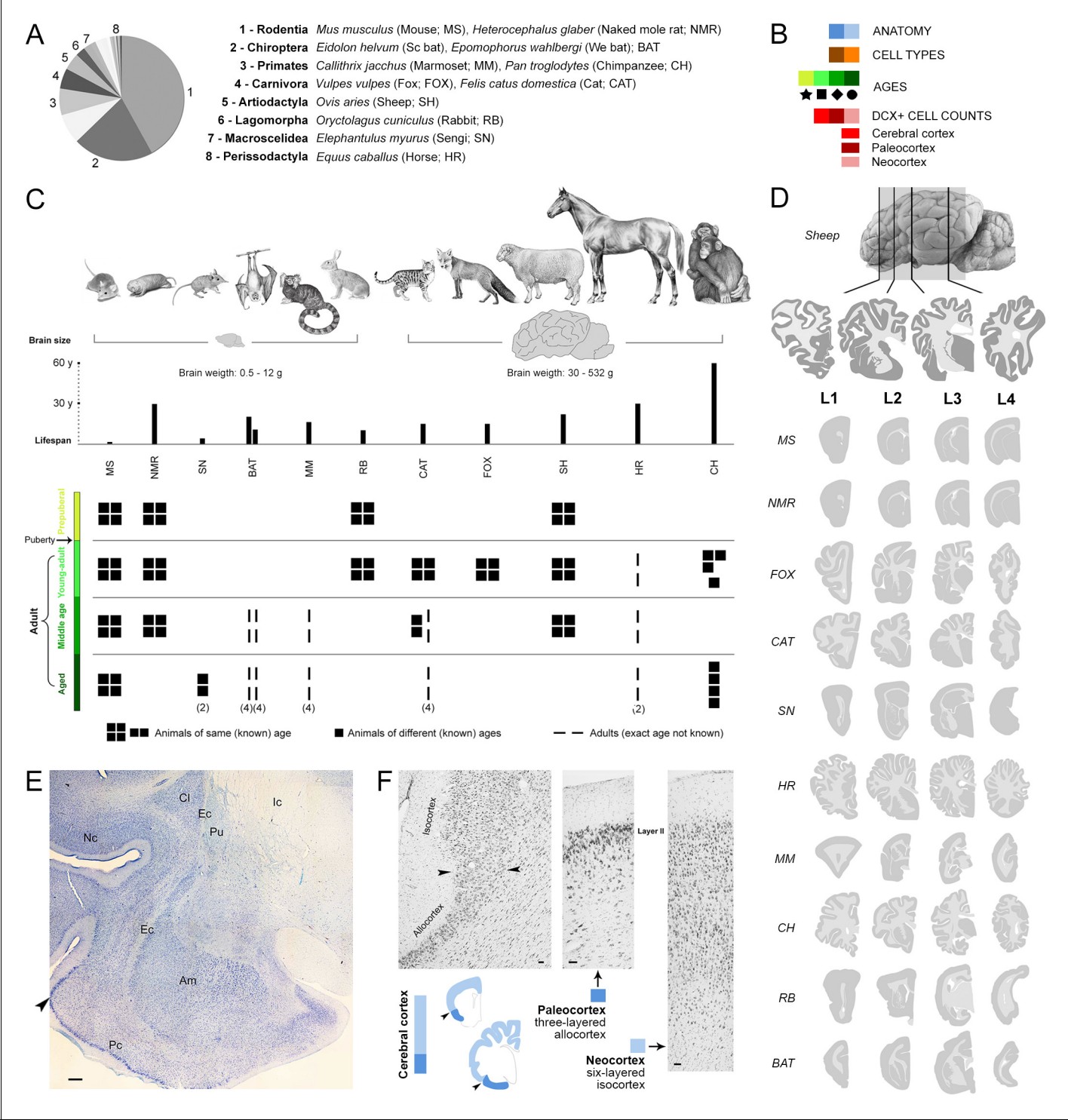

**Figure 1.** Sample of species, ages, and comparable brain regions (coronal levels) of the mammals used in this study (*Supplementary file 1*). (**A, C**) Mammalian species and orders (scientific name, common name – used hereafter – and abbreviation) with special reference to their brain size and lifespan (C, top). (**B**) Color code (see also *Figure 4*). (C, bottom), Different ages considered for each species; all groups are composed of four individuals (apart from different specification; *Supplementary file 1*). (**D**) Four anterior-posterior brain levels, identified by comparable neuroanatomy based on histology (example of a cat brain section in E), and correspondent mini-atlases for each animal species; representative sections within the thickness of each level are shown. (**F**) Paleocortex-neocortex transition (arrowheads), identified histologically as a shift from three to six cortical layers. Scale bars: E, 500 μm; F, 50 μm.

The online version of this article includes the following figure supplement(s) for figure 1:

*Figure 1 continued on next page*

*Figure 1 continued*

**Figure supplement 1.** Summary of the qualitative and quantitative analyses performed in this study.

dendrites; *Figure 2D,E*; *Piumatti et al., 2018*; *Bonfanti and Nacher, 2012*; *König et al., 2016*). The cell soma size range was 2–9 µm in diameter for type 1 cells, and 7–17 µm for type 2 cells (ranges for each species in *Figure 2E*).

Since type 2 cells are known to be less immature than type 1 (*Piumatti et al., 2018*; *Rotheneichner et al., 2018*), each was counted separately (*Figure 2D*). In all mammals considered, type 1 cells were more abundant than type 2 cells, the latter representing 3–16% of the total DCX+ cells (apart from rodents, *Figure 2E*). Among rodents, mice showed the highest percentage of type 2 cells (around 40%) and naked mole rats the lowest (1–4%) (*Figure 2E*).

On the whole, morphology and cell type proportions of the layer II DCX+ neurons were rather constant in the different species and ages studied, whereas their anatomical distribution in the neocortex was variable across phylogeny.

## Cell proliferation and degree of immaturity/maturity

We examined staining for markers of cell proliferation (Ki-67 antigen; PCNA in bats) in adult cerebral cortex samples from each brain level, with particular reference to layer II and DCX+ neurons. In rabbit and sheep, bromodeoxyuridine (BrdU) was injected in adult animals to determine whether DCX+ cells could have been generated in adulthood. The periventricular (SVZ) and hippocampal (SGZ) neurogenic zones were used as internal, positive controls (*Figure 3A*). No DCX/Ki-67 or DCX/BrdU double-labelled cells were ever observed in the cortex of any of the species investigated (*Figure 3A*). Only rare Ki-67, PCNA or BrdU immunoreactive nuclei were detectable in the cortex (with an average number of proliferating cells from 0 to 5 cells for the cortical area considered per cryostat section analysed), never in association with DCX+ cells (*Figure 3A* and *Figure 3—figure supplement 1*). These proliferating nuclei were identified as glial cells in double staining carried out for astrocytic and oligodendrocytic glial markers (*Figure 3A*; *Selinfreund et al., 1991*; *Boda et al., 2015*).

Two additional markers were used in association with DCX to assess the neuronal maturational stage of the cINs in mouse, cat, rabbit, and marmoset (*Figure 3B*; *Piumatti et al., 2018*): PSA-NCAM, a low-adhesive form of N-CAM widely present in neurons during the development of the nervous system (*Bonfanti, 2006*) and expressed by cells retaining plasticity during adulthood (*Hoffman et al., 1982*; *Bonfanti et al., 1992*; *Bonfanti and Nacher, 2012*); NeuN, an RNA-binding protein expressed by postmitotic neurons that start differentiation (*Mullen et al., 1992*), which can identify most types of mature neurons with some exceptions (*Gusel'nikova and Korzhevskiy, 2015*), and is expressed in type 2 cINs (*Piumatti et al., 2018*; referred to as 'complex cells' in *Rotheneichner et al., 2018*).

Across the mammalian species in our sample, only 17–18% of the DCX+ cells co-expressed NeuN, with the NeuN+/DCX+ neurons (*Figure 3C*) mostly characterized by the type 2 cell morphology, in accord with previous observation in mouse and sheep (*Piumatti et al., 2018*; *Rotheneichner et al., 2018*). About 14–39% of the DCX+ cells were immunoreactive for PSA-NCAM (with no particular relation with cell morphology; *Figure 3C*).

## Quantitative analysis of cortical layer II DCX+ cells across mammals

We assessed the numbers of DCX+ cells per mm of layer II perimeter (linear density: calculated in the cerebral cortex, in paleocortex and neocortex; *Figure 4—figure supplement 1B*), at the four brain levels (L1–L4) at the ages shown in *Figure 1C* (total number of brains analysed = 80). Linear density was measured since cINs are arranged in a monolayer-like row within cortical layer II (hosting all DCX+ cells of the cortex). Such density, calculated on the real cortical layer II length measured in entire brain coronal sections (highly varying in different species and ages), represents a comparable value, allowing inferences across different mammals. The total count of DCX+ cells was performed in each coronal section (total number of sections = 960) from both paleocortex and neocortex, in order: i) to establish in each species, the exact anatomical location of the cINs in the cortex (considering two extremes, in chimpanzees, a total of 1774.25 cm of cortical layer II were evaluated with an average of 18.88 cm for each cryostat section, while in mice, 167.14 cm of cortical layer II were

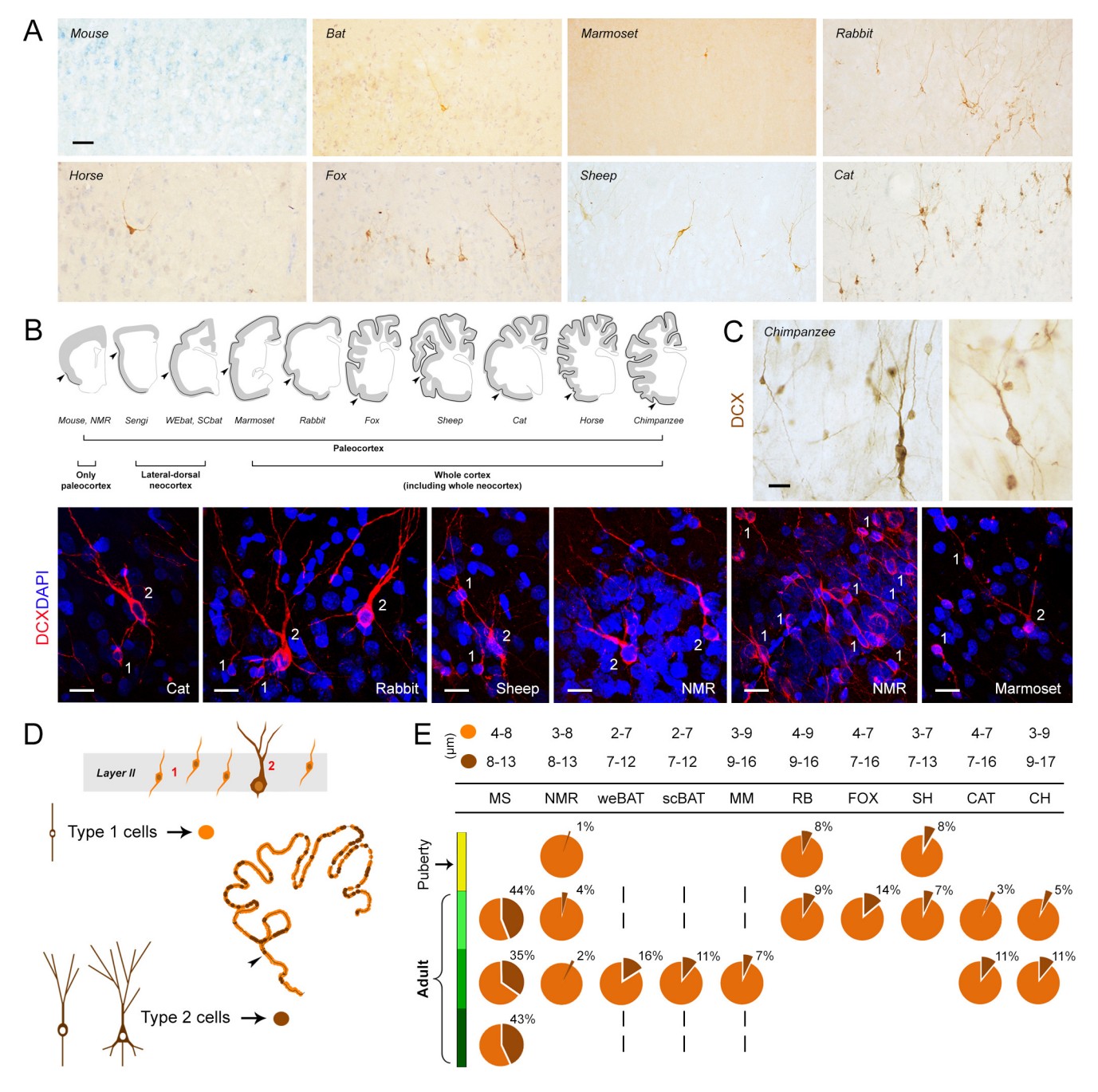

**Figure 2.** Occurrence, regional distribution and cell types of layer II DCX+ cells in the mammalian cerebral cortex. (**A**) Representative images of the DCX+ neurons in neocortex. Scale bar: 50 μm. (**B**) Extension of cINs (DCX+ cells) in the cortical layer II (black line); arrowheads: paleocortex-neocortex transition. (**C,D**) Cell types and morphology of cINs; (**C**) representative examples of type 1/type 2 cortical DCX+ neurons in different mammalian species; scale bars: 20 μm; (**D**) Counting of type 1 and type 2 cells (pie charts showing the percentages in **E**, bottom). (**E**, top) Cell soma diameters (expressed as min/max range) of the DCX+ cortical cells in each species.

The online version of this article includes the following figure supplement(s) for figure 2:

**Figure supplement 1.** Representative images of DCX+ neurons in the paleocortex of different mammal species.

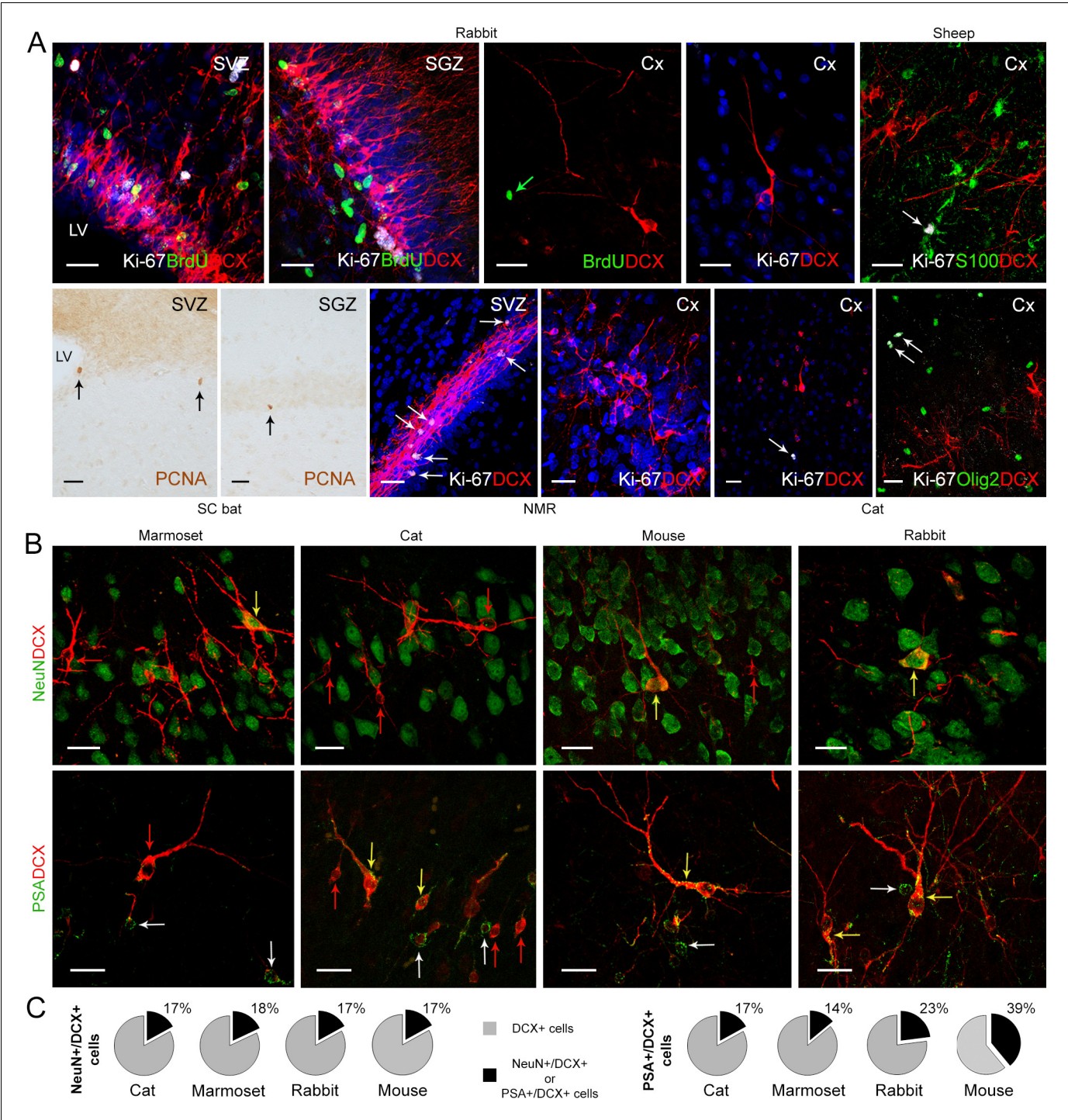

**Figure 3.** Detection of markers for cell division and neuronal maturity/immaturity in cortical layer II of adult mammals. (**A**) Cell proliferation visualized in neurogenic subventricular zone (SVZ) and subgranular zone (SGZ), here used as an internal control, with nuclear markers of local cell proliferation. None of these markers were detectable in association with DCX+ immature neurons in the cortical layer II; some isolated Ki-67+ nuclei occasionally present in cortex do not co-label with DCX, likely corresponding to glial cells (see double staining with S-100β, top right, and Olig2, bottom right). (**B**) Subpopulations of DCX+ immature neurons in the cortical layer II co-express NeuN (mostly type 2 cells) and PSA-NCAM (both cell types); red arrows, single-stained DCX+ cells; white arrows, single-stained PSA-NCAM+ cells; yellow arrows, double-stained cells. (**C**) Percentage of NeuN+ and PSA-NCAM+ cells among the DCX+ cells. Scale bars: 25 μm.

The online version of this article includes the following figure supplement(s) for figure 3:

*Figure 3 continued on next page*

*Figure 3 continued*

**Figure supplement 1.** Quantification of proliferating cells in upper neocortical layers (I, II, III) in mouse, NMR, marmoset, rabbit, sheep and cat considering Ki-67+ cells/mm$^2$ (PCNA+ cells/mm$^2$ and BrdU+ cells/mm$^2$ were considered in SC bat and rabbit, respectively).

evaluated with an average of 0.86 cm for each cryostat section; ii) to identify each cell as belonging to either type 1 or type 2 morphology (total number of cells counted = 414.008; *Figure 2D* and *Figure 1—figure supplement 1*). The number of layer II cortical DCX+ cells was investigated in all species for which a brain hemisphere was available from 4 individuals (n = 10 species; qualitative analysis only was performed on two additional species: sengis and horses).

By comparing pre-puberal specimens to adult ones, in rodents a dramatic decrease in the linear density was found (nonparametric Mann Whitney test, p<0.01; data referred to cerebral cortex; *Figure 4A*). In rabbit, a slight age-related decrease was observed (nonparametric Mann Whitney test, p<0.05), whereas in sheep no significant differences were detectable between the two age groups. These data suggest that maturation of the cINs, while rapidly occurring at young ages in rodents (thus eroding the reservoir of immature cells), progressively slows in a larger brained mammal, the sheep, leaving a greater remaining population of immature neurons in the adult. To define the abundance of this cINs reservoir, we quantified the cortical layer II DCX+ cells in adult specimens from a diverse group of species (pooled across three stages: young-adult, middle age and aged; *Figure 1C*). A high degree of interspecific variability in the density of cINs was found in cerebral cortex, with notable differences between paleocortex and neocortex (*Figure 4B*). When data were organized in two groups (*Figure 1C* and *Supplementary file 2*), to compare small (brain size range 0,5 to 12 g) to large brains (brain size range 30 to 384 g; which also have more neocortical surface; *Supplementary file 3*), a significant higher density was observed in the latter group with respect to the former (nonparametric Mann Whitney test, p<0.0001; *Figure 4—figure supplement 2A*). When considering the neocortex and paleocortex separately, the same trend in DCX+ cell linear densities was observed (see *Figure 4C,D* and *Figure 4—figure supplement 2B,C*). The linear density in paleocortex was higher than in neocortex for all species (a significant correlation in densities from paleo and neocortex was present - nonparametric Spearman correlation, p<0.001; *Figure 4—figure supplement 3*); variation among species is more evident in neocortex (median linear density in neocortex in cat is 17.58 cells/mm and in mouse 0, while in paleocortex in cat is 58.41 cells/mm and in mouse 1.18 cells/mm; *Figure 4C* and *Figure 4—figure supplement 2B*).

To investigate if the occurrence of cINs might be linked to other species-specific factors, the linear density of DCX+ cells in cerebral cortex was correlated with encephalization quotient and lifespan, but no correlations were found (not shown). To investigate variance in density of DCX+ neurons in relation to brain weight, neocortical surface area, layer II perimeter, brain length, we performed a Principal Component Analysis (PCA). The first principal component explained 78% of the variance and the second principal component explained 19% of the variance. In particular, measures of brain size contributed more to the loading of the first component, whereas density of DCX+ neurons loaded most on the second component, showing that species grouped according to these biological features (*Figure 5B* and Discussion). To determine the scaling relationships in our datasets, phylogenetic generalized least squares (PGLS) regression analysis was performed. There was a significant relationship between linear density of DCX+ neurons and species mean brain weight (adjusted $r^2$ = 0.618, p=0.02), with a moderate phylogenetic signal (Pagel's lambda = 0.24) (*Figure 5D*).

There was also a significant relationship between DCX+ neuron density and layer II perimeter as measured from the same brains in our sample (adjusted $r^2$ = 0.595, p=0.03, Pagel's lambda = 0.00). The relationship between DCX+ neuron density and gyrification index approached significance (adjusted $r^2$ = 0.328, p=0.10, Pagel's lambda = 0.00; *Figure 5—figure supplement 1*).

Finally, to determine whether the distribution of DCX+ immature neurons in cortical layer II might be heterogeneous through the anterior-posterior extension of the brain, the neocortical linear densities obtained in the four coronal levels (L1-L4) were compared in each species. Two-way ANOVA with Bonferroni post-hoc tests found no significant differences among brain levels in any species, except for cats (lower cell density in L4: L1 vs L4: p<0.001; L2 vs L4: p<0.001; L3 vs L4: p<0.01; *Figure 4E*). In a heatmap analysis, animals belonging to the same orders (chimpanzee and marmoset, fox and cat, mouse and NMR) were clustered together (*Figure 4F*).

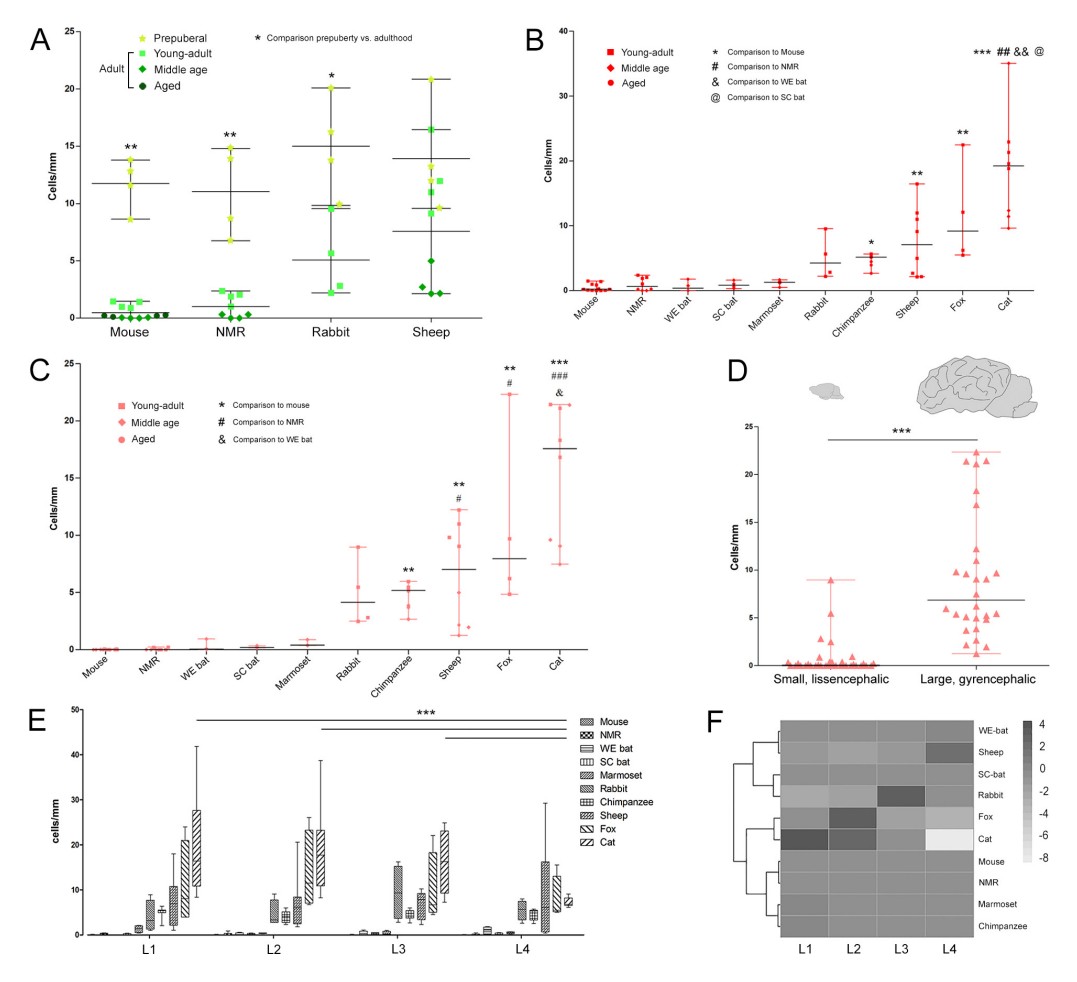

**Figure 4.** Quantification of DCX+ cINs density in the mammalian cortex. (**A**) Linear density and statistical analysis of DCX+ cells in the cerebral cortex of four mammalian species at different ages (see color code in **Figure 1B**); a sharp drop between prepuberal and adult ages is detectable in rodents but not in sheep. (**B–D**) DCX+ cell linear densities and statistical analysis referred to the total extension of the cerebral cortex (**B**) and to the neocortex (**C**) of 10 mammalian species (adult specimens considered – see color code in **Figure 1B**); a high heterogeneity is detectable from mice to cat, even more evident when grouping small, lissencephalic and large, gyrencephalic mammals (**D**). (**E**) No main differences are detectable in the distribution/amount of cINs among different anterior-posterior brain levels (except for cat); (**F**) heatmap of the distribution of linear density in different brain levels (L1-L4; neocortex).

The online version of this article includes the following figure supplement(s) for figure 4:

**Figure supplement 1.** Quantification of DCX+ cells (linear densities in the cortical layer II).

**Figure supplement 2.** Quantification of DCX+ cINs density in the mammalian cortex.

**Figure supplement 3.** Correlation study between DCX+ cell density in paleo- and neo-cortex.

## Discussion

We systematically explored phylogenetic patterns in the cIN population of mammalian brains. Our results indicate that, in contrast to the rodents examined, there is a trend for larger, more gyrencephalic brains to contain higher densities of these cells across the entire neocortical mantle. The substantial homogeneity of features found for the cIN population in each species, including its occurrence in the paleocortex of all species, the type 1 and type 2 cell morphologies and their relative percentages, the staining of additional markers for immaturity (including PSA-NCAM, a very fragile antigen as consisting of a carbohydrate exposed on the extracellular portion of N-CAM; **Bonfanti, 2006**) and cell proliferation, and the consistency of results after checking for markers within the internal controls (see Results, summary in **Figure 5A**, left, and discussion of cell densities below),

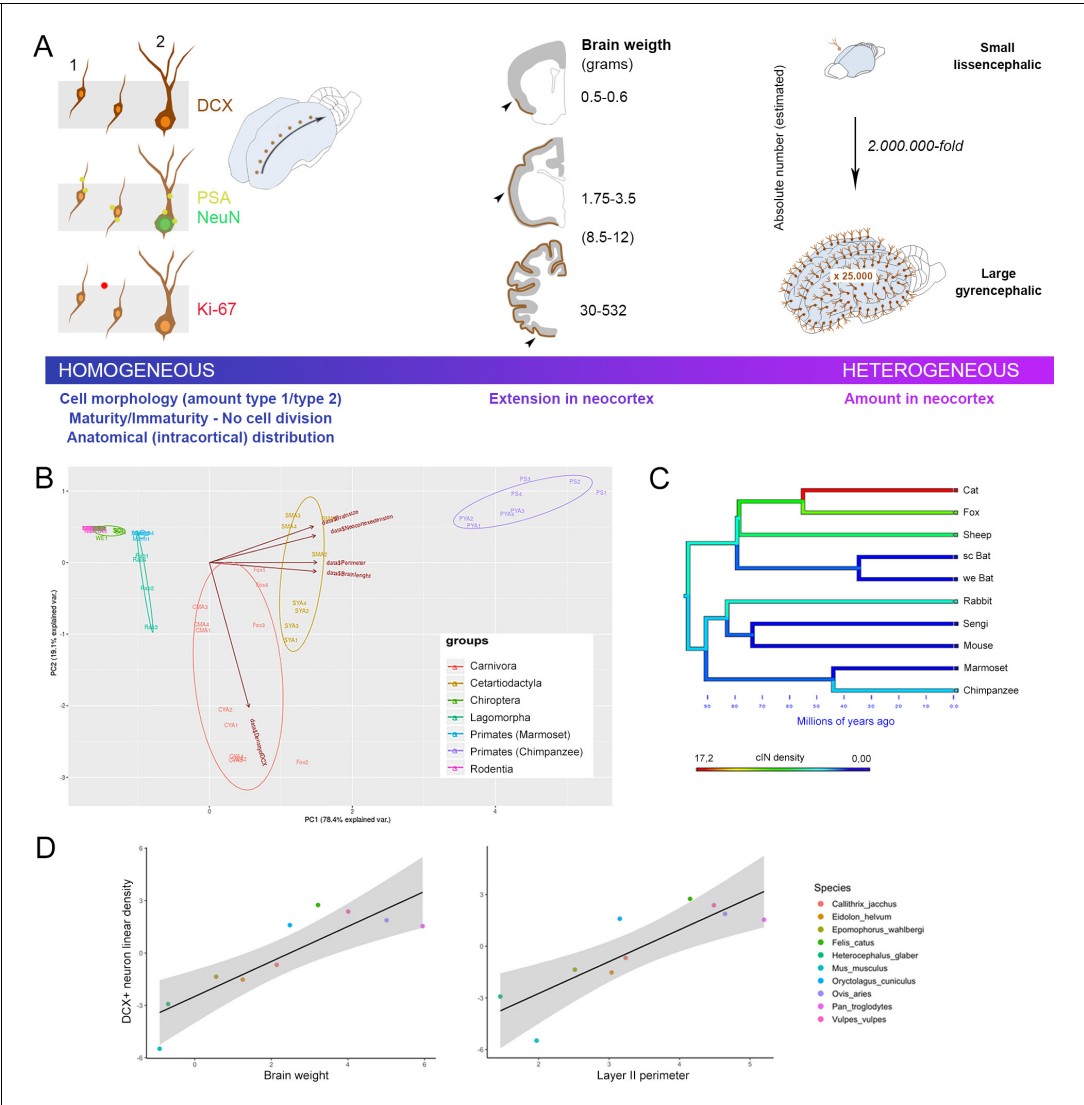

**Figure 5.** General features of cortical immature neurons are highly preserved in mammals whereas their amount is greater in large-brained species. (**A**) Some aspects, including morphology, cell types, degree of immaturity and non-proliferative activity (left), appear quite constant across the mammalian species, whereas their extension in the neocortex (middle) and their overall amount (right) vary remarkably, increasing from small, lissencephalic to large, gyrencephalic brains. On the right, the estimation of total number of cINs in mouse and chimpanzee calculated by multiplying the median number of DCX+ cells in the neocortical layer II for the number of cryostat sections cut from the entire hemispheres; **Supplementary file 4**). (**B**) Principal Component Analysis (PCA; the different species are arranged according their orders; see text). The animal species are identified by different colours and abbreviations: CYA for Cat Young-Adult, CMA for Cat Middle Age, SYA for Sheep Young-Adult, SMA for Sheep Middle Age, PYA for Primates Young-Adult Chimpanzee, PS for Primate (Senior) aged Chimpanzee. (**C**) Map of character evolution on the phylogenetic tree illustrating the independent emergence of neocortical DCX+ neuron densities in the mammalian species considered. (**D**) PGLS regression showing that linear density of neocortical DCX+ neurons covaries significantly with brain weight and layer II perimeter.

The online version of this article includes the following figure supplement(s) for figure 5:

**Figure supplement 1.** PGLS regression analysis between layer II DCX+ neuron density and gyrification index.

confirm that the remarkable differences found in the anatomical distribution and amount (linear density) of this neuronal population are real interspecies differences, linked to phylogenetic variation rather than technical issues.

## Large-brained mammals possess a far larger reservoir of cINs in their neocortex

The linear density found in large-brained mammals (chimpanzee, fox, sheep, cat) was as much as one order of magnitude higher in comparison to the species with small, lissencephalic brains (with the only exception of rabbits, which are known to display unexplained high levels of structural plasticity; *Luzzati et al., 2003*; *Luzzati et al., 2006*; *Ponti et al., 2008*). Such an increase is far more evident in the neocortex: if considering an estimation of the absolute number of cINs, a nearly 2 million-fold difference emerges between mouse and chimpanzee (*Figure 5A*, right, and *Supplementary file 4*). The residual variance in these relationships may be due to differences in various factors that we were not able to control in the sample, such as rearing history or lifestyle of the animals (e.g., captivity or wild). Our current results, nevertheless, demonstrate that mammals show considerable variability in numbers of cINs across their cerebral cortices and that densities tend to generally be associated with brain size. The finding of a greater immature neuron population present in the neocortex of some large-brained mammals reveals a reservoir of undifferentiated cells in an expanded brain region characterized by higher computational capacities (*Roth and Dicke, 2005*; *Roth, 2015*; *Zilles et al., 2013*). The occurrence of more inter-individual variations in large-brained, gyrencephalic mammals suggests that having a greater reservoir of cINs might favor the possibility of their modulation through life. Future studies focused on single species involving groups of animals kept in different, highly controlled, environmental conditions for long periods of time are required to reveal if external factors can modulate the population (reservoir) of cINs in individuals.

No correlations emerged by considering lifespan or categorical aspects of ecological specializations, such as habitat or dietary preferences. Considering parameters linked to brain size and allometric scaling, PCA confirmed that species tend to be clustered along two main axes of variance that separate on the basis of brain size measures and cIN density, respectively (*Figure 5B*). In addition, PGLS regression analysis indicates that linear density of DCX+ neurons covaried with brain weight, layer II perimeter, and gyrification index (*Figure 5D*). The fact that cINs appear to increase significantly in association with mammalian brain size among phylogenetically divergent taxa such as primates, artiodactyls, and carnivores suggests the independent evolution of this phenotype.

## The general features and intracortical distribution of cINs are quite constant in all species

Aside from their cortical distribution (in terms of anatomical distribution within the entire neocortex of each species) and density, other features of the cINs investigated here (morphology, occurrence and relative amount of cell types, degree of maturity/immaturity, non-proliferative state) were substantially similar regardless of the species considered (*Figure 5A*, left). Even the spatial distribution within the cortex (considering the anterior-posterior brain levels in the neocortex of each species); *Figure 5A*, left), showed no substantial variation, thus indicating that there is not likely a strong link between occurrence of cINs and specific functional cortical areas. In a heatmap analysis cats were the only species to show minor differences among anatomical levels that we sampled in the brain, the disparity being limited to a drop in cell density in the occipital region (*Figure 4E,F*). Finally, no substantial differences in cIN density and distribution were found by comparing species with five-layered (e.g., sheep, which lacks layer IV and is generally accompanied by expansion of layers II and III; *Cozzi et al., 2017*), and six-layered cortex (the remaining species). Accordingly, supragranular layer II (but not layer IV) persists through the evolution of the mammalian brain independently of the organization of the cortex in five or six layers (*Cozzi et al., 2017*).

On the whole, the analyses carried out in the present study indicate that cINs are a cell population with a set of phylogenetically conserved features independent from cortical anatomy and its functional specializations, yet, with increased distribution in mammals with enlarged neocortices. Our results strongly suggests that this trait and the mechanisms and processes that are underpinning it has evolved independently several times in mammals (with convergent gains and losses). Our sampling regime encompassed both large and small brained species in three of the four eutherian mammal superorders: Laurasiatheria, Euarchonoglires and Afrotheria, thought to have initially diverged in the Cretaceous Period (*Nishihara et al., 2009*). As such, it would be of great interest to establish whether common pathways have been selected to produce the phenotypes we observe.

## Implications for cortical structural plasticity

Whether and how the abundance of cINs in large-brained mammals could be linked to cortical function and higher-order cognitive abilities merits further investigation. The complex relationship between expansion of the brain and the increase of computational capabilities remains to be understood (*Healy and Rowe, 2007*). Nevertheless, the overall phylogenetic distribution of cINs suggests reconsideration of the mechanisms of plasticity in large-brained mammals. The mammalian brain has low capacity for cell renewal, with most neurons being lifelong, mature elements. The exception represented by adult stem cell niche-depending neurogenesis is spatially restricted (*Bond et al., 2015*; *Bao and Song, 2018*; *Forrest et al., 2018*; *Lim and Alvarez-Buylla, 2016*), does not serve the neocortex and is thought to be reduced in large-brained species (*Sanai et al., 2011*; *Paredes et al., 2016*), the issue being still debated for the hippocampus (*Kempermann et al., 2018*; *Petrik and Encinas, 2019*). The cINs, as a special type of undifferentiated cells generated before birth but retaining molecular profiles of immaturity (*Bonfanti and Nacher, 2012*; *König et al., 2016*), share features with very young, highly plastic neurons which are still capable of remarkable structural changes (i.e. newborn neurons; *Bonfanti and Nacher, 2012*; *Brown et al., 2003*). We suggest that cINs may serve as an important reservoir of undifferentiated cells in large-brained mammals. The relative occurrence of elements showing higher or lesser degrees of immaturity on the basis of their morphology and cell marker expression (as previously described by *Piumatti et al., 2018*; *Rotheneichner et al., 2018*; *Benedetti et al., 2020*), further supports this view: type 2 cells (the less immature elements; *Piumatti et al., 2018*; *Rotheneichner et al., 2018*) were abundant in mice (animals with short lifespan and fast metabolism), whereas type 1 were more prevalent in naked mole rats (neotenic mammals retaining features of immaturity for their extended lifespan; *Penz et al., 2015*); all other species retain substantial amount of type 1 cells even at advanced ages. Type 1 cells, as highly immature neurons (*Piumatti et al., 2018*; *Rotheneichner et al., 2018*; *Benedetti et al., 2020*), might retain a phenotype that permits a form of plasticity in large expanded (relatively 'stable') neocortices in terms of disposable undifferentiated cells. In that sense, cINs should not be considered as an alternative to canonical neurogenesis, rather a parallel form of plasticity providing undifferentiated neurons, in the absence of cell division, in a region of the mammalian brain of utmost importance for cognition, not endowed with much capacity for neurogenesis in adulthood. According to a recent theory on the origin of the neocortex (*Aboitiz and Montiel, 2015*), neurogenesis persists in evolutionary 'old parts' of the mammalian brain linked to olfaction (archicortex: olfactory bulb and hippocampus), which were of paramount importance in ancient mammals. These systems were subsequently replaced/integrated by the expansion of the isocortex as a 'multimodal interface' for behavioral navigation based on other sensory modalities (vision and audition) recruited into the expanding neocortex and contributing to multimodal association networks (*Aboitiz and Montiel, 2015*). As a result, larger mammalian brains are more composed of cortex, ranging from under 20% in relative volume in rodents to over 80% in humans (*Hofman, 1989*). In this evolutionary context, the cINs might represent an option for providing a reservoir of undifferentiated neurons in a brain structure not served by adult neurogenesis.

It is poorly understood why these cells are restricted to layer II. Superficial layers (II and III) are involved in integrating corticocortical information and in associative learning, with respect to deep cortical layers mainly linked to subcortical structures. Accordingly, the proportion of cortex they occupy is largest in primate species and smallest in rodents, indicating difference in importance devoted to corticocortical connectivity across mammals (*Hutsler et al., 2005*). In addition, neocortical histological organization develops in a sequence with pyramidal neurons of the deepest layers generated first and neurons exiting the stem cell pool later migrating to the more superficial layers (*McConnell, 1995*), a feature possibly linked to an extended maturational time of large brains. Finally, under the profile of their evolutionary origin, upper layers of the neocortex are thought to come through co-option of the olfactory cortex (*Luzzati, 2015*). On these bases, superficial layers might be more suitable to retain a reservoir of undifferentiated, plastic cells with respect to lower layers more specialized to host extracortical projection neurons.

In conclusion, during evolution, the expanded neocortices of large-brained species might have adopted cINs as a reservoir of young cells compatible with their substantial stability and reduced capacity for neurogenesis, independent from singular functional specializations. The persistence of a significant population of immature cells could be part of a neocortical architecture shared by

phylogenetically divergent species as a developmental correlate of brain enlargement. The immature neuron population revealed here might represent one of the anatomical substrates of the so called 'brain reserve' or 'cognitive reserve' which is thought to allow the maintenance of efficient cognitive functions throughout life and to exert a protective effect against aging (*Stern, 2017*; *La Rosa et al., 2019*). Accordingly, the cIN population might be also important in the progressive maturation of cortical circuitries during postnatal and young ages. The occurrence and distribution of cINs in humans, as well as their possible modulation by physiological and/or environmental conditions in animal models or postmortem human brain tissue, are worthwhile of further investigation. The prevalent occurrence of such a reservoir in neocortices of large-brained mammals will represent a great challenge for future studies.

## Materials and methods

### Brain sample

Brains used in this study were collected from various institutions and tissue banks, all provided by the necessary authorizations (see below and *Supplementary file 1*). All experiments were conducted in accordance with current EU and Italian laws.

Four prepuberal, four young adult, four middle age and four aged mice were analysed. Perfusion was performed under anesthesia (i.p. injection of a mixture of ketamine, 100 mg/kg, Ketavet, Bayern, Leverkusen, Germany; xylazine, 5 mg/kg; Rompun, Bayer, Milan, Italy; authorization of the Italian Ministry of Health and the Bioethical Committee of the University of Turin; code 1112/2016-PR - courtesy of Annalisa Buffo) and brains were postfixed for 4 hr.

Four prepuberal naked mole rat brains were extracted a few minutes following euthanasia in accordance with Schedule 1 of the Animals (Scientific Procedures) Act 1986, and fixed by immersion, while four young-adult and four middle age specimens were perfused. All brains were postfixed overnight.

Four prepuberal and four young-adult female rabbits were used. Rabbits received one daily injection of BrdU (Sigma; 40 mg/Kg) for 5 consecutive days and then were killed 10 days after the last injection. Animals were deeply anesthetized (ketamine 100 mg/kg - Ketavet, -and xylazine 33 mg/kg body weight - Rompun) and perfused with fixative (Italian Ministry of Health, authorization n. 66/99-A). Tissues were postfixed for 6 hr.

WE bats were captured in Nairobi, Kenya, and SC bats were captured in Kampala, Uganda. Permits and licenses were granted by the National Museums of Kenya and the Uganda National Council of Science and Technology (No. 024/07/1). Animals were trapped during night and kept in cages for 1–3 days before perfusion. They were deeply anaesthetized with sodium pentobarbital (Nembutal, 60 mg/ml; 50 mg/kg) and perfused. Brains were removed and postfixed for 2–18 days. The actual ages of the animals are not known, but they were aged as adults on the basis of the following criteria: the closure of the femoral and humeral epiphyseal plate, the body weight, the forearm length and sexual maturity (evidence of lactation or pregnancy in female and testis size in male; *Gatome et al., 2010*).

Eastern rock sengis were caught in Sherman life traps at the Goro Game Reserve, Limpopo Province, South Africa (Permit 0089-CPM-401–00004, CITES and Permit Management Office, Department of Environ- mental Affairs, Limpopo Province). Tissues were harvested from animals euthanized under projects in accord with the ethics guide-lines of South Africa (University of Pretoria Clearance EC028-07) and the guidelines of the American Society of Mammalogists. Animals were trapped during night and, the next day, they were deeply anesthetized with pentobarbital (50 mg/kg), perfused and post-fixed for 24 hr (*Slomianka et al., 2013*).

Marmoset brains were extracted 1 hr after death and post-fixed for 3 months. The exact ages of the animals are unknown; they were aged as adults by experienced veterinarians, as described for bats.

The foxes were euthanized with 5% sodium-thiopental, decapitated and perfused with fixative. Brains were dissected and postfixed in changes of fixative for 3–7 days. Experiments were conducted following the international guiding principles for biomedical research involving animals developed by the Council for International Organizations of Medical Sciences (CIOMS) and were also in compliance with the laws, regulations, and policies of the 'Animal welfare assurance for humane care

and use of laboratory animals,' permit number A5761-01 approved by the Office of Laboratory Animal Welfare (OLAW) of the National Institutes of Health, USA (*Huang et al., 2015*).

Four prepuberal and four young-adult sheep were perfused. The brains were dissected out, cut into blocks and post-fixed in the same fixative for 48 hr. Two years old animals received four intravenous injections of BrdU (1 injection/day, 20 mg/Kg in 0.9% saline; Sigma-Aldrich, France; *Piumatti et al., 2018*; *Brus et al., 2013*). Three different survival times were analyzed: 1, 2 and 4 months (maturation time for neuroblasts in sheep is 1–4 months; *Brus et al., 2013*). Four middle age animal brains were collected 20 min after death, fixed and kept in fixative for 1 month.

Four young-adult and four middle age cats, and two adult horse brains were extracted post-mortem (the PMI was less than 1 hr for cats and between 2 and 20 min for horses), fixed and kept in the fixative solution for a 1 month (cat) and 3 months (horse).

Four young-adult and four aged chimpanzee brains from the National Chimpanzee Brain Resource (www.chimpanzeebrain.org) were used. Within 14 hr of each subject's death (body refrigerated soon after death), the brain was removed, immersed in 10% formalin and fixed for 10–14 days. The specimens were collected post-mortem from zoos and primate research centers, maintained in accordance with each institution's animal care guidelines (*Schenker et al., 2010*).

## Tissue processing for histology and immunohistochemistry

The whole brain hemispheres were cut into coronal slabs (1–2 cm thick). The slabs were washed in a phosphate buffer (PB) 0.1 M solution, pH 7.4, for 24–72 hr (on the basis of brain size) and then cryoprotected in sucrose solutions of gradually increasing concentration up to 30% in PB 0.1 M. Then they were frozen by immersion in liquid nitrogen-chilled isopentane at −80°C. Before sectioning, they were kept at −20°C for at least 5 hr (time depending on the basis of brain size) and then cut into 40 µm thick coronal sections using a cryostat or a sliding microtome. Free-floating sections were then collected and stored in cryoprotectant solution at −20°C until staining.

Sections were used for histological staining procedures and immunocytochemistry. Histological analyses were performed on Toluidine blue - or cresyl violet - stained sections. For 3,3'-diaminobenzidine (DAB) immunohistochemistry, sections were rinsed in PBS 0.01 M, pH 7.4. Antigen retrieval was performed using citric acid at 90°C for 5 min. After further washing in PBS 0.01 M, pH 7.4, the sections were immersed in appropriate blocking solution (1–3% Bovine Serum Albumin, 2% Normal Horse Serum, 0,2–2%Triton X-100 in 0.01 M PBS, pH 7.4) for 90 min at RT. Following, sections were incubated with primary antibodies (see below) for 48 hr at 4°C. After washing in PBS 0.01 M, pH 7.4, sections were incubated for 2 hr at RT with secondary antibodies (Anti-goat, made in horse, 1:250 - BA-9500; Anti-goat made in rabbit, 1:250 – BA-5000; Anti-rabbit, made in horse, 1:250 – BA-1100; Anti-mouse made in horse, 1:250 – BA-2001; Vector Laboratories, Burlingame, CA 94010). Then, sections were washed with PBS 0.01 M, pH 7.4 and incubated in avidin–biotin–peroxidase complex (Vectastain ABC Elite kit; Vector Laboratories, Burlingame, CA 94010) for 1 hr at RT. The reaction was detected with DAB, as chromogen, in TRIS-HCl 50 mM, pH 7.6, containing 0.025% hydrogen peroxide for few minutes and then washed in PBS 0.01 M, pH 7.4. Sections were counterstained with Cresyl violet, mounted with DPX Mountant (Sigma-Aldrich, 06522) and coverslipped.

For immunofluorescence staining, sections were rinsed in PBS 0.01 M, pH 7.4. Antigen retrieval was performed using citric acid at 90°C for 20 min. After further washes in PBS 0.01 M, pH 7.4, sections were immersed in appropriate blocking solution (1–3% Bovine Serum Albumin, 2% Normal Donkey Serum, 1–2% Triton X-100 in 0.01M PBS, pH 7.4) for 90 min at RT. Then the sections were incubated for 48 hr at 4°C with primary antibodies (see below), and subsequently with appropriate solutions of secondary antibodies for 4 hr at RT: Alexa 488-conjugated anti-mouse (1:400; Jackson ImmunoResearch, West Grove, PA - 715-545-150), Alexa 488-conjugated anti-rat (1:400; Jackson ImmunoResearch, West Grove, PA - 712-546-153), Alexa 488-conjugated anti-rabbit (1:400; Jackson ImmunoResearch, West Grove, PA - 711-545-152), Cyanine 3 (Cy3)-conjugated anti-goat (1:400; Jackson ImmunoResearch, West Grove, PA - 705-165-147), Alexa 647-conjugated anti-mouse (1:400; Jackson ImmunoResearch, West Grove, PA - 715-605-151). Immunostained sections were counterstained with 4',6-diamidino-2-phenylindole (DAPI, 1:1000, KPL, Gaithersburg, Maryland USA) and mounted with MOWIOL 4–88 (Calbiochem, Lajolla,CA).

Primary antibodies and dilutions used for this study: goat anti-DCX (1:500–3500, Santa Cruz Biotechnology, Santa Cruz, CA - sc-8066), mouse anti-Ki-67 (1:500–1000, BD Pharmigen - 550609), rabbit anti-Ki-67 (1:600–1000, Leica-Novocastra - NCLKi67p), rat anti-BrdU (1:300, AbDSerotec,

Kidlington, UK - OBT0030), mouse anti-PCNA (1:30000, Delta Biolabs, Gilroy, Calif - DB095), mouse anti-PSA-NCAM (1:1400, Millipore, Bellerica, MA - MAB5324), mouse anti-NeuN (1:300, Millipore, Bellerica, MA - MAB377), rabbit anti-S100β (1:5000, Swant, Swiss Antibodies, CH - 37A); rabbit anti-Olig2 (1:1000, Millipore, Bellerica, MA - AB9610).

## Comparable neuroanatomy

The mammalian brains included in the current study differ in terms of brain size, gyrencephaly and overall neuroanatomical organization. In order to perform comparable analyses, four correspondent anterior-posterior, coronal brain levels (L1-L4) were designated (*Figure 2B*). To find the neuroanatomical landmarks of these brain levels in each species, the entire hemispheres were cut into 40 μm thick coronal sections. Then the corresponding levels were defined as coronal (thick) slices involving the same main brain structures: L1, from anterior opening of the lateral ventricle (or shift from lateral to olfactory ventricle in rabbit and sheep) to L2; L2, from anterior starting of internal capsule to L3; L3, from anterior starting of the claustrum and/or amygdala to L4; L4, from posterior closing of the lateral ventricle to an extension equivalent to that of L2 (same number of 40 μm thick serial sections).

For species in which neuroanatomy was previously described, existing atlases were used as a reference (mouse: *Allen Institute for Brain Science*; marmoset, chimpanzee, rabbit, fox, cat: *Comparative Mammalian Brain Collections*; sheep: *Michigan State University*) then matching the brain levels on our specimens; for the remaining species the procedure was performed by using our histologic specimens (example given in *Figure 2E*). The paleocortex-neocortex (allocortex-isocortex) transition was easily identifiable on the cresyl violet-stained sections (*Figure 2F*).

## Image acquisition, processing and data analysis

Images were collected using a Nikon Eclipse 90i microscope (Nikon, Melville, NY) connected to a color CCD Camera, a Leica TCS SP5, Leica Microsystems, Wetzlar, Germany and a Nikon Eclipse 90i confocal microscope (Nikon, Melville, NY). Quantitative analyses were performed using Neurolucida software (MicroBrightfield, Colchester, VT) on DCX-DAB stained sections. The linear density of DCX + cells present in layer II of the cerebral cortex was evaluated in 4 individuals of 10 species (in each brain level, three sections were considered – one in the anterior, one in the central and one in the posterior part). In each section, the total perimeter of cortical layer II was traced and all DCX+ cells along its length were counted (linear density = number of cells/mm). In addition to the linear density for the cortex, also separate densities for neocortex and paleocortex were evaluated. In the same sections, the morphology of cINs was evaluated and the number of type 1 and type 2 cells was counted using different markers selected from the 'markers toolbar' in Neurolucida software.

The cell soma size (diameter) was obtained by evaluating the width orthogonal to main axis, measured in about 100 cells for each animal species using the Neurolucida 'measure line' tool.

In cat, sheep, rabbit, marmoset, NMR and mouse, the number of Ki-67+ nuclei (PCNA in SC bat) and the number of Ki67+/DCX+ double-stained cells were counted in an area corresponding to the neocortical upper layers (I, II, III; Ki-67+ cells/mm$^2$). For each specimen, a cryostat section from each of the 4 brain levels was selected and, in each of them, three microscopic fields (40x magnification; 1 dorsal, 1 lateral, 1 medial along neocortical extension) were analyzed. In rabbits, the same analysis was also performed for BrdU+ nuclei and BrdU+/DCX+ double-stained cells.

The percentage of NeuN+/DCX+ and PSA-NCAM/DCX+ double-stained cells was calculated in cat, rabbit, marmoset and mouse by analyzing eight microscopic fields (40x magnification; 2 dorsal, 2 lateral, 2 ventral; 2 medial in the cortical perimeter) from a central section in each of the four levels for each specimen.

All images were processed using Adobe Photoshop CS4 (Adobe Systems, San Jose, CA) and ImageJ version 1.50b (Wayne Rasband, Research Services Branch, National Institute of Mental Health, Bethesda, Maryland, USA). Adjustments to color, contrast, and brightness were made.

## Statistical analysis

All graphs and statistical analyses were performed using GraphPad Prism Software (San Diego California, USA) using different nonparametric tests: Mann-Whitney test, Kruskal-Wallis test with Dunn's multiple comparison post test, Two-way ANOVA with Bonferroni post-hoc test and Spearman

correlation coefficient r. p<0.05 was considered as statistically significant. Median was used as a central measure. Free software environment R was used to calculate and draw heatmaps and perform PCA. In particular for the heatmap, in order to highlight similarities and differences in the linear densities distributions of INs among the four neocortical layers across species, density values were transformed by subtracting the mean value of each species, and *pheatmap* package (*Kolde, 2019*) was used with *correlation* as parameter value for row clustering distance and *column* as parameter value for *scale* function argument and PCA was calculated by prcomp package (*Vq, 2011*) with center and scale parameters set to TRUE.

The species median DCX+ neuron densities (neocortex) were used to perform an ancestral character state reconstruction of trait evolution mapped onto the phylogeny. This was implemented in Mesquite software, using a parsimony model. To determine the scaling relationships in our dataset we employed PGLS regression with a likelihood-fitted lambda transformation. The species median DCX+ neuron linear densities (neocortex) as the main variable of interest were used in these analyses. The PGLS was run against three different predictors - brain weight, layer II perimeter, and gyrification index. All data were log transformed prior to PGLS to fit power functions to linear regression, as is standard procedure. A phylogenetic tree of the species in the sample was downloaded from the TimeTree database (*Kumar et al., 2017*). All regression plots are on a log scale and show the 95% confidence intervals.

## Acknowledgements

The present work was supported by MIUR-PRIN2015 (grant 2015Y5W9YP) and University of Turin (PhD program in Veterinary Sciences). The National Chimpanzee Brain Resource is supported by NIH grant NS092988. We thank Frederic Lévy (INRA, Nouzilly, France), and Annalisa Buffo (NICO, Orbassano, Italy) for sheep and mouse brains, respectively, and Roberta Parolisi, Marco Ghibaudi, Elaine Miller, and Cheryl Stimpson for laboratory assistance.

## Additional information

### Funding

| Funder | Grant reference number | Author |
|---|---|---|
| Ministero dell'Istruzione, dell'Università e della Ricerca | 2015Y5W9YP | Luca Bonfanti |
| NIH Blueprint for Neuroscience Research | NS092988 | Chet C Sherwood |
| University of Turin | PhD program in Veterinary Sciences | Chiara La Rosa |

The funders had no role in study design, data collection and interpretation, or the decision to submit the work for publication.

### Author contributions

Chiara La Rosa, Data curation, Formal analysis, Investigation, Methodology, Writing - original draft, Writing - review and editing; Francesca Cavallo, Alessandra Pecora, Formal analysis; Matteo Chincarini, Chris G Faulkes, Juan Nacher, Resources; Ugo Ala, Software, Formal analysis; Bruno Cozzi, Resources, Funding acquisition, Methodology, Writing - review and editing; Chet C Sherwood, Resources, Formal analysis, Funding acquisition, Writing - original draft, Writing - review and editing; Irmgard Amrein, Resources, Methodology, Writing - review and editing; Luca Bonfanti, Conceptualization, Resources, Supervision, Funding acquisition, Investigation, Visualization, Methodology, Writing - original draft, Writing - review and editing

### Author ORCIDs

Matteo Chincarini http://orcid.org/0000-0001-6369-4992
Luca Bonfanti https://orcid.org/0000-0002-1469-8898

## Ethics

Animal experimentation: All experiments were conducted in accordance with current laws regulating experimentation in each contry/institution providing the brain tissues used in this study. Mice: authorization of the Italian Ministry of Health and the Bioethical Committee of the University of Turin; code 1112/2016-PR - courtesy of Annalisa Buffo. NMR: Naked mole-rats were maintained in the Biological Services Unit at the Queen Mary University of London. Because tissue sample collection was post-euthanasia, in full accordance with National (Schedule 1 of the Animals - Scientific Procedures - Act 1986) and Institutional animal care and use guidelines, additional local ethical approval for NMR work was not required for this study. Rabbits: Italian Ministry of Health, authorization n. 66/99-A WE bats and SC bats: Permits and licenses from the National Museums of Kenya and the Uganda National Council of Science and Technology (No. 024/07/1) Sengis: CITES and Permit Management Office, Department of Environmental Affairs, Limpopo Province (permit 0089-CPM-401-00004); ethics guidelines of South Africa (University of Pretoria Clearance EC028-07) Marmosets: Animal handling and tissue collection was conducted according to the Animal Welfare Act (AniWA) of the Federal food safety and veterinary office, Switzerland. Foxes: Council for International Organizations of Medical Sciences (CIOMS), also in compliance with the laws, regulations, and policies of the "Animal welfare assurance for humane care and use of laboratory animals," permit number A5761-01 approved by the Office of Laboratory Animal Welfare (OLAW) of the National Institutes of Health, USA. Sheep: Animal care and experimental treatments complied with the guidelines of the French Ministry of Agriculture for animal experimentation and European regulations on animal experimentation (86/609/EEC) and were performed in accordance with the local animal regulation (authorization No. 006352 of the French Ministry of Agriculture in accordance with EEC directive). Ewes were killed by a licensed butcher in an official slaughterhouse (authorization No. A37801). Aged sheep and horses: the brains were collected at a local slaughterhouse. Slaughtering of horses and other species raised for meat and/or milk production are treated according to the European Community Council directive (86/609/EEC) concerning animal welfare during the commercial slaughtering process, and constantly monitored under mandatory official veterinary medical care. Cats: The brains of the cats were removed during routine diagnostic post-mortem and therefore no ethical permission is required. Chimpanzee: all experimental procedures with postmortem chimpanzee tissue were carried out according to the National Institutes of Health guidelines for animal research and were approved by the Institutional Animal Care and Use Committee at the George Washington University (IACUC A117).

## Decision letter and Author response

Decision letter https://doi.org/10.7554/eLife.55456.sa1
Author response https://doi.org/10.7554/eLife.55456.sa2

# Additional files

## Supplementary files

• Supplementary file 1. Animals used in this study.

• Supplementary file 2. Main information on the animal species considered in this study. Species with lissencephalic, small-brains are shown in regular font and species with gyrencephalic, large-brains are shown in italics.

• Supplementary file 3. Estimation of the neocortex surface area (calculated by using the median length of the layer II perimeter multiplied for 40 µm - thickness of sections - for the number of sections of the entire hemisphere).

• Supplementary file 4. Estimation of the total number of DCX+ cells in layer II of the neocortex (calculated by multiplying the median number of DCX+ cells per section for the number of sections). Immature neurons prefer large brains

• Transparent reporting form

## Data availability

All data generated or analysed during this study are included in the manuscript and supporting files. For information regarding ages and socio-ecological data we used the Animal Diversity Web: Myers P, Espinosa R, Parr CS, Jones T, Hammond GS, Dewey TA. 2019. The Animal Diversity Web. Available at: https://animaldiversity.org. Animal Diversity Web. This is also cited in the Reference list of the manuscript.

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
