## [Decision Letter]

**Acceptance summary:**

Your work focuses on the immature neuron population in cortical layer II and discusses differences in occurrence, distribution and fate between different mammalian species ranging from small-lissencephalic to large-gyrencephalic brains. While these immature cells are virtually absent in rodents, they are present in the entire neocortex of many other species, where their density appears to vary with brain size. Given the great current interest in adult hippocampal neurogenesis (a debate recently reviewed in Behav Brain Res 2020), your carefully performed study on this rare material advances our insight in a relatively unknown, 'other' population of immature cells and does so from a broader, evolutionary perspective that will interest the field.

**Decision letter after peer review:**

Thank you for submitting your article "Phylogenetic variation in cortical layer II immature neuron reservoir of mammals" for consideration by *eLife*. Your article has been reviewed by three peer reviewers, one of whom is a member of our Board of Reviewing Editors, and the evaluation has been overseen by Catherine Dulac as the Senior Editor.

The reviewers have discussed the reviews with one another and the Reviewing Editor has drafted this decision to help you prepare a revised submission.

Summary:

The paper provides a new evolutionary perspective on immature (stem-cell like) cells that they find to be virtually absent in rodents, yet are present in the entire neocortex of many other species, in close relation to brain size.

As carefully performed, timely study on rare material, it addresses an important topic that is closely related to adult neurogenesis, and provides an important contribution to our understanding of structural plasticity across species, a topic that has come into the spotlight in recent years due to conflicting studies about human neurogenesis. This manuscript advances our insight in a relatively unknown population of immature, 'plastic' cells from an important broader, evolutionary perspective.

Reviewer #1:

La Rosa et al. focus on the immature neuron population in cortical layer II and discuss differences in occurrence, distribution and fate between different mammalian species ranging from small-lissencephalic to large-gyrencephalic brains.

Their compiled data show these immature cells to be virtually absent in rodents yet present in the entire neocortex of many other species, where their density appears to vary with brain size; they find larger, more gyrencephalic brains to contain higher densities of these cells across the entire neocortical mantle.

This is rare material and a carefully performed, descriptive study on a timely topic that has received a lot of attention lately in view of the debate on human neurogenesis following the Sorrells paper in Nature 2018. I found the manuscript very well written and altogether a solid, comprehensive paper that advances our insight in this relatively unknown population of immature, 'plastic' cells from a broader, evolutionary perspective.

I have the following comments;

1) In the subsection “Characterization of DCX+ cortical neurons across mammals”, the remark on the role of fixation and conditions is very short, but an important, critical aspect of such comparative studies, as highlighted by the discussions and commentaries following the Sorrells 2018 paper. The importance of these methodological aspects cannot be underestimated also for these tissues and requires more discussion. Given e.g. the huge spread in cells/mm e.g. in Figure 4D, can they exclude that such methodological factors, like fixation time, postmortem delay etc., have contributed? Or to variation between species?

2) The same applies to influences of stress, early life/rearing/environmental conditions (see also Chawana et al., 2020 and discussions in Kempermann et al., 2018) in general that can at least lastingly modify neurogenesis in the hippocampus, and could potentially also modify cIN cells.

3) In view of the important effect of post-capture stress on DCX signal, e.g. in bats (Chawana et al., 2014), how long after capture were e.g. the bat, sengis and fox brains collected? Did the marmoset, cat and chimp brains also have a postmortem interval?

4) Rather than solely studying the number, or developmental stage (ki/DCX/PSA-NCAM) of the cINs, also their activational state, and to what extent these cells can be recruited by environmental challenges, could differ between species. As others have e.g. done for hippocampal neurogenesis in relation to (spatial or fear-related) learning, have the authors considered or tried co-labeling their BrdU, or DCX cells, for immediate early genes like c-fos or Zif268?

5) If the cIN cells, as the authors suggest, constitute a 'reservoir' of disposable undifferentiated cells, when do they think this reservoir will be used and upon which stimuli would they become involved? Do they suspect responses under e.g. conditions of cortical damage?

6) Given the emerging interest in closely related plasticity changes in other brain regions like e.g. the hypothalamus and amygdala and other species (see e.g. Ernst/Frisen/Marlatt/Benier/Parent/Dayer/Cameron/Kokoeva/Pierce/Xu/Paul/Francis/Blackshaw), for an overall perspective, a brief description of this topic, citing these papers, will be quite relevant.

7) In relation, they mention it briefly in the Discussion, but what would be a functional role that could be attributed, and how could this be tested in future studies, to (the cIN population in) this particular cortical layer?

8) Aside from the size argument, I do not understand why the authors think that '... the cortical superficial layers might reasonably be the best place to retain a reservoir of undifferentiated, plastic cells.' Please explain further.

9) When quantifying cell numbers per linear density, this was mostly done in 3 anatomical levels for each species; were anatomical differences within the structure apparent? Given the clear differences between e.g. (neurogenesis in) rostral and caudal parts of the hippocampus, some discussion on this would be informative.

Reviewer #2:

In this interesting, unique study, La Rosa and colleagues follow an interesting lead, suggesting the existence of long-lived immature neurons in the neocortex of mammals, especially gyrencephalic species. The team of Luca Bonfanti has been pioneering these investigations and the hypothesis fits well with concepts of "neurogenesis without division" etc. and the general idea that for neurogenesis in a wider sense, the interval between division of the precursor cell and final maturation of the new neuron might be very long. Such neurons would provide a reservoir for life-long cellular in the absence of real adult neurogenesis (involving the entire process from division to maturation and integration).

La Rosa and colleagues provide excellent immunohistochemical evidence of immature cortical neurons. One has to follow the argument, however, that DCX-expression, partly together with NeuN or PSA-NCAM is a sufficient argument for immaturity and hence the nature of this interesting population of cells. This conclusion is largely drawn by analogy from mostly the author's own work from piriform cortex. If one accepts this premise, this is a very interesting study.

1) The authors should make a clear statement, why no absolute cell counts could be obtained and that relative numbers have certain limitations. These must be discussed for the analyses provided. As with aging the reference volume might variably change between and within species, comparisons are semi-quantitative at best. For the core type of conclusion to be drawn here, this is not much of an issue, but it must be made clear. Furthermore, early in the text the concept of linear density must be explained to avoid misunderstandings.

2) The coverage of the literature is rather selective and quite biased to a limited set of authors. Other studies about layer II neurogenesis (disputed or not) are not discussed (e.g. Bifari et al., 2017). The entire discussion appears to be deliberately somewhat detached from the rest of the field. This is rather strange given the fact that the work by Luzzati, Amrein and others makes important contributions to the key discussions and is widely appreciated. The authors should integrate their work with the emerging ideas in the entire field, including its controversies.

3) Issue #1 notwithstanding, the quantitative conclusions are not fully clear. With the extension of layer 2, the number of the immature neurons appears to increase. What would that mean?

Reviewer #3:

The authors perform a systematic quantification of immature-like DCX+ cells in the cortex of a variety of mammals. Their primary finding is that there is substantial variability in the density of these cells, with more cells present in larger-brained animals. They propose that these DCX+ cells are generated prior to adulthood but retain immature features, to compensate for lower rates of neurogenesis in large-brained animals. Overall, this is an interesting paper which I think will make an important contribution to our understanding of structural plasticity across species, a topic that has come into the spotlight in recent years due to conflicting studies about human neurogenesis. I don’t have any major concerns.

---

## [Author Response]

Reviewer #1:[…] I have the following comments;1) In the subsection “Characterization of DCX+ cortical neurons across mammals”, the remark on the role of fixation and conditions is very short, but an important, critical aspect of such comparative studies, as highlighted by the discussions and commentaries following the Sorrells 2018 paper. The importance of these methodological aspects cannot be underestimated also for these tissues and requires more discussion. Given e.g. the huge spread in cells/mm e.g. in Figure 4D, can they exclude that such methodological factors, like fixation time, postmortem delay etc., have contributed? Or to variation between species?

We are aware of some aspects of heterogeneity concerning the treatment of the brain material coming from animal species which, for practical or ethical reasons, cannot be raised in laboratories and then perfused to fix the brain rapidly (post-mortem intervals, type and duration of fixation). These aspects cannot be made completely uniform when addressing, at the same time, diverse species of different origin and endowed with different brain size, namely, the aim of our study.

We studied 84 brains (80 used for quantifications) to have a wide representation of mammals; the reviewer him/herself speaks of “rare material”, and we thank him/her for recognizing this. To analyse these highly different animal species in the most possibly comparable way we had to accept some differences in the procedures.

On these premises, all efforts were made to ensure quality control. The post-mortem interval, the most critical parameter, was generally very low, as reported in the present revision.

In particular, we trust our results concerning the different amount of DCX+ cells in different species as not being affected by methodological factors for several reasons:

A) multiple aspects of the cINs which were investigated here were constantly present in our immunocytochemical assay, in all species analysed:

i) the morphological aspect of the DCX+ cells, including the type 1 and type 2 cells and their relative percentages;

ii) the good quality of the staining;

iii) the occurrence of the staining in the internal controls (both neurogenic zones in the same animals);

iv) the staining of other markers (PSA-NCAM, NeuN) and their relative percentage of co-expression with DCX. The detection of an antigen as fragile as the PSA-NCAM is (namely, a carbohydrate) further supports the quality of our brain tissues.

In addition to the main goal of our study (to investigate the occurrence and amount of cINs in widely different mammals), the careful, multifaceted analysis of the above mentioned aspects allowed us to characterize and find the same “system” represented by occurrence of the cIN population in the layer II showing very similar features in all species (mostly expected after the previous reports in the paleocortex of laboratory rodents).

B) the animal species in which the cINs appear very abundant are those with the larger brains, namely those which were not perfused; by contrast, some of the species which were perfused (hence, no post-mortem interval delay; e.g., mouse, naked mole rat) showed a low density. Again, this fact indicates that the striking differences in cIN amount are not linked to fixation.

C) our study is carried out in the cerebral cortex, namely, a superficial part of the brain which is usually fixed rapidly even by immersion.

Taken together, all these aspects indicate that differences in the spatial distribution and amount of cINs are not linked to technical issues, but to actual neurobiological differences among species (the above listed answers A, B and C can also be used for points 2 and 3).

Changes: We tried to summarize these concepts in the text (especially in the subsection “Characterization of DCX+ cortical neurons across mammals”, where indicated by the reviewer), since we agree with the reviewer that they were underdeveloped. The post-mortem intervals were added in the text and in Supplementary file 1 (as requested in point 3) and further discussion on this issue has been introduced at the beginning of the Discussion section.

2) The same applies to influences of stress, early life/rearing/environmental conditions (see also Chawana et al., 2020 and discussions in Kempermann et al., 2018) in general that can at least lastingly modify neurogenesis in the hippocampus, and could potentially also modify cIN cells.

As discussed in the previous point for fixation, also stress, early life/rearing/environmental conditions might exert an effect in modulating cINs as they do on adult neurogenesis, nevertheless:

– Both Chawana’ and Kempermann’ papers are referring to the hippocampus, in which adult neurogenesis (a different process with respect to immature neurons, involving cell proliferation) is known to be highly modifiable by various environmental conditions. We do not know at present if the same conditions can affect cINs, which are different from newly born neurons, since generated before birth.

As to the possible effects of these aspects on our results please refer to the answers to Point 1 (A, B, C);

Note: a very interesting finding in the Chawana paper is that the occurrence (amount) of newly born neurons in the hippocampus of wild bats is rather stable, as if genetically determined in the species living in that particular environment. This is in accord with previous studies suggesting that environmental modulation, though particularly potent in standardized/genetically identical, laboratory rodents, is far less evident in wild living species (reviewed in Amrein, 2015). This view seems to fit well with our findings of species-specific amount of cINs in different mammals.

– The crucial methodological factor identified by Kempermann et al., 2018 (for humans) is postmortem delay. In humans, stressful environmental conditions antemortem, such as hospitalization, respiratory illness or coma, are well known to affect RNA quality analyzed in brain tissue postmortem (see for example Durrenberger et al., J Neuropath & Exp Neurol 69, p.70 (2010)). It could well be that such factors lay at the core of the highly debated adult neurogenesis in humans, but none of these antemortem factors apply to the material analyzed here. None of the animals were sick, all were rapidly killed, brains dissected and tissue was fixed according to brain size;

Changes: This aspect is briefly mentioned at the beginning of the second paragraph in the Results. Though referring to hippocampus (but here given as a general example), the references of Chawana et al., 2020 and Kempermann et al., 2018 were added. As discussed in point 4, some aspects related to the possible modulation of cINs, though interesting, are, at present, too speculative and should be addressed deeply in future studies focused on cIN modulation in well-defined experimental conditions (maybe in single animal species).

3) In view of the important effect of post-capture stress on DCX signal, e.g. in bats (Chawana et al., 2014), how long after capture were e.g. the bat, sengis and fox brains collected? Did the marmoset, cat and chimp brains also have a postmortem interval?

Taking into account that we were dealing with individuals from species that are not typically found in laboratories, some of which were captured in the wild, the PMI were generally very short.

The foxes were farm-breed; as described in the Materials and methods section, they were euthanized and perfused (no PMI). Also marmosets were from zoos, thus brains were collected immediately after retrieving the animals from their enclosure, with a PMI of 1 hour. The postmortem delay was limited to a few minutes for the sheep (only for middle age sheep, because prepuberal and young animals were perfused so there is no PMI) and the horse, as the brains of those two species were collected immediately after slaughtering in a controlled environment. The cat brains were collected during postmortem procedures performed within one hour after death.

As to the time between capture and death of bats and sengis: bat brains were collected between 1 and 3 days after trapping, sengi around 12 hours. Again, concerning our results, we refer to the answers to point 1A, B, C (in particular: DCX staining was clear and strong in the piriform cortex of sengis, as shown in Figure 2—figure supplement 1).

Changes: Some integrations relative to this point were made in the text (Materials and methods). The missing post-mortem intervals (PMI) have been added in the text and in Supplementary file 1.

4) Rather than solely studying the number, or developmental stage (ki/DCX/PSA-NCAM) of the cINs, also their activational state, and to what extent these cells can be recruited by environmental challenges, could differ between species. As others have e.g. done for hippocampal neurogenesis in relation to (spatial or fear-related) learning, have the authors considered or tried co-labeling their BrdU, or DCX cells, for immediate early genes like c-fos or Zif268?

Usually, in experimental designs in which the neural activation is investigated using IEGs, the animals are exposed to specific stimuli (e.g., ultrasound vocalizations to stimulate auditory and limbic systems; Ouda et al., 2016) and sacrificed immediately after the end of the stimulation (maximum 30 minutes later), considering the short life of these markers. Animals used in this study lived in different conditions (laboratory or wild) and did not receive any concordant stimulus in the time window preceding death: in this scenario, it is impossible to obtain sensible data regarding the activational state of cINs using IEGs.

Apart from the above considerations, since the first decade of March (some days before we received the reviewer’s comments) Italy was already in forced lockdown for Covid-19 emergency. Since then, we had no possibility to go to our laboratories at NICO research center and University (apart from animal care and clinical tasks that cannot be postponed). We are currently starting to experience new rules for re-entry, in shifts of half-day, twice a week.

Changes: We expanded the discussion of such issues, by adding some relevant references, but we tried to keep them within limits, because we think that at present they would be too speculative.

5) If the cIN cells, as the authors suggest, constitute a 'reservoir' of disposable undifferentiated cells, when do they think this reservoir will be used and upon which stimuli would they become involved? Do they suspect responses under e.g. conditions of cortical damage?

We do not know, at present, which might be the role/usefulness of a cortical “reservoir” of immature cells.

By long-lasting experience in brain structural plasticity, we think that mammals have generally low capacity for brain repair, and that most plasticity is used for learning/maintaining an efficient and plastic brain during life. This is the reason why we think that cINs can be useful in the “brain reserve” rather than for brain repair.

Changes: We added the hypothesis of brain reserve in the Discussion, as we suggested in a previous review article, but, again, this is at present highly speculative, though surely interesting.

General comment to points 2, 4 and 5:

We agree with the reviewer on the importance and interest of exploring the possible modulation of the cIN population based on different environmental conditions or experimental procedures. Nevertheless, the present study is aimed at considering many animal species in search for phylogenetic trends, while the above mentioned aspects could be surely addressed in future by specific studies (focused also on methodological approaches such as Chawana et al., 2020) carried out on a single animal species in which different animal groups can be controlled.

We think that showing cIN modulation would not involve only searching for possible differences in their number (an aspect which might show no variations, considering the fact that this population is generated before birth) but to investigate deeper features, such as for instance the expression of synaptic markers and the fine modifications in the complexity of dendritic arborisation. We foresee that laboratories with experience in comparative neuroanatomy and wide sources of unconventional animal species will be pivotal in understanding if and how cINs can be modulated.

6) Given the emerging interest in closely related plasticity changes in other brain regions like e.g. the hypothalamus and amygdala and other species (see e.g. Ernst/Frisen/Marlatt/Benier/Parent/Dayer/Cameron/Kokoeva/Pierce/Xu/Paul/Francis/Blackshaw), for an overall perspective, a brief description of this topic, citing these papers, will be quite relevant.

Due to the complexity of this issue and the extensive literature accumulated in brain structural plasticity, we think that these two topics should be kept separated to maintain the focus of interpretation directly on our results.

As to the neurogenesis issue we agree with several reviewers that it is not directly relevant to this work, so we reduced our discussion of it so as not to be a digression.

As to the issue of different types of plasticity, it is becoming more and more evident that they can be differently adopted by different species. We can synthetize the current knowledge as follows: beside the extremes represented by cINs (non-newly generated) and newly born neurons of the neurogenic niches (SVZ, hippocampus, hypothalamus), the emerging results/hypotheses in the field suggest that in certain brain regions (such as amygdala) non-newly generated immature neurons and newly generated neurons could coexist, the newly born ones being more prevalent during young ages (Marlatt et al., 2011; Sorrells et al., 2019; focusing on primate and human amygdala). Yet, the data referring to subcortical regions are still fragmentary and many questions are still open due to a lack of systematic results (comparable results in different species and ages), so that it is difficult to establish if neurogenesis is present or not in such regions. For these reasons, we think it is too early to address such an issue here and we choose to focus on the cerebral cortex and immature neurons.

Changes: As reported in other points/answers (e.g., raised by reviewers #2 and 3), we reduced extended discussion about adult neurogenesis, keeping the text more focused on the cerebral cortex, especially in the Discussion.

We inserted the references of Dayer (interneuron genesis in the mouse cortex).

The image in Figure 5E was deleted.

7) In relation, they mention it briefly in the Discussion, but what would be a functional role that could be attributed, and how could this be tested in future studies, to (the cIN population in) this particular cortical layer?

The assessment of possible modulatory effects on the cINs from different environmental/experimental conditions is an extremely interesting subject, but it is outside the objective of the present work. By revealing the occurrence of cIN reservoir in large-brained mammals, our work opens a wide range of future possible studies. To address these latter, it will be necessary to establish different animal groups belonging to the same species, including controls, to test different conditions, and then perform deep analyses which cannot be limited to the cell count but maybe involve careful analysis of the complexity of their dendritic arborizations. This could be a very long study, even on a single animal species.

Moreover, considering our data, it could be more useful to try this kind of approach on a mammalian species endowed with a more consistent reservoir of cINs, extended to the whole neocortex, to test if specific tasks, for example, could modify cINs (in number and/or in maturational stages) in certain functional areas. This will be no easy task.

Changes: We strengthened in the Discussion the fact that our findings open many new questions and possibilities (we agree with the reviewer that they were underdeveloped and we made some suggestions) but we tried to avoid too many speculations.

8) Aside from the size argument, I do not understand why the authors think that '... the cortical superficial layers might reasonably be the best place to retain a reservoir of undifferentiated, plastic cells.' Please explain further.

On the basis of current knowledge, it is not easy to formulate a hypothesis explaining why a population of immature neurons should be restricted to one cortical layer and why, specifically, to cortical layer II. We deeply explored the existing literature but we did not found solid explanations for the specific functions of cortical layers, especially for the most superficial ones (II and III). However, it is interesting that this reservoir is hosted in the supragranular layer II that persists through the evolution of the mammalian brain independently of the organization of the cortex in five or six layers, is the latter to be generated during cortical layer development, and is thought to come through co-option of the olfactory cortex.

Changes: We modified and re-wrote the relative sentences in the Discussion, hoping that now this might be clearer. We also added the above mentioned concepts as current hypothesis.

9) When quantifying cell numbers per linear density, this was mostly done in 3 anatomical levels for each species; were anatomical differences within the structure apparent? Given the clear differences between e.g. (neurogenesis in) rostral and caudal parts of the hippocampus, some discussion on this would be informative.

The quantifications were performed on 4 brain levels (3 is the number of coronal sections considered for quantification at each level), identified in each species by the presence of the same neuroanatomical structures independently from the different brain sizes. Since also another reviewer asked for further explanation of the counting method, we realized that it was not well enough described in the manuscript, so we tried to better explain it and the reasons why it was chosen.

Changes: We added a new figure (now Figure 1—figure supplement 1) summarizing all the analyses carried out in our work, and an image in Figure 4—figure supplement 1, with a more detailed explanation in the Figure legend.

As suggested by the reviewer, one of the first questions we asked ourselves during this study was whether the cINs might be more present or abundant in specific areas/functional domains of the cortex. One stimulating hypothesis was that animal species with different cortical specializations would have different occurrence/amount/distribution of cINs. Nevertheless, a surprising and clear result of our analysis is the rather constant presence of the DCX+ cells, appearing homogeneously distributed in the different anatomical levels or brain areas of each species. Apart from the striking difference in total amount of immature neurons among species, in each species, in the whole cortical portions in which they exist they are present/distributed homogeneously. This substantial homogeneity was observed in all species, in all anterior-posterior locations (brain levels). These distributions are confirmed by the graphic and heatmap in Figure 4E, F.

Changes: We created paragraph subdivisions in the Discussion and one of them is now: “The general features and intracortical distribution of cINs are quite constant in all species”. We strengthened the concept of an “inherent/constitutive package” of cINs that is possessed with very similar features (but in highly different amount and anatomical distribution) by the animal species.

As a link with point 1 of reviewer #2, and as an answer to that point, this is one of the reasons we choose to count *all* DCX+ cells in the *entire* neocortical II perimeter, in order to check if differences in presence/density were detectable in different tracts (cortical areas). The new supplementary figure (now Figure 1—figure supplement 1) should contribute to better explain this.

Reviewer #2:[…] La Rosa and colleagues provide excellent immunohistochemical evidence of immature cortical neurons. One has to follow the argument, however, that DCX-expression, partly together with NeuN or PSA-NCAM is a sufficient argument for immaturity and hence the nature of this interesting population of cells. This conclusion is largely drawn by analogy from mostly the author's own work from piriform cortex. If one accepts this premise, this is a very interesting study.

We thank the reviewer for appreciating our work. We used the expression “neurogenesis without division” provided by the reviewer him/herself, which explains well the concept underlying the cINs.

As to the nature of these cells as plastic elements, there are now several solid papers performed on the mouse piriform cortex, not only coming from our laboratory.

Very recently, a group from the Paracelsus Medical University in Austria used a transgenic mouse in which the DCX+ cells in the piriform cortex can be visualized with GFP and followed through time (Rotheneichner et al., 2018; Benedetti et al., 2019) to show that cINs do mature through the age of the animal and can integrate into the layer II.

In 2008, Juan Nacher showed that these cells are generated during development (before birth) then remaining an immature state during adulthood (Gomez-Climent et al., 2008); we recently showed that the layer II cINs are generated before birth also in the neocortex and in a gyrencephalic mammal (the sheep), by using BrdU treatment during embryonic development and its subsequent detection in newborn lambs (Piumatti et al., 2018).

The occurrence and features of these cells are now well known, though important questions were open:

i) whether they are differently present/distributed in phylogeny;

ii) whether they can be abundantly present in the whole neocortex. The present work was addressing such questions and the results show that they are not just a curious cell type confined within the paleocortex.

1) The authors should make a clear statement, why no absolute cell counts could be obtained and that relative numbers have certain limitations. These must be discussed for the analyses provided. As with aging the reference volume might variably change between and within species, comparisons are semi-quantitative at best. For the core type of conclusion to be drawn here, this is not much of an issue, but it must be made clear. Furthermore, early in the text the concept of linear density must be explained to avoid misunderstandings.

We think that our explanation of the analyses carried out in our study was not clear enough and complete in the manuscript. Hence, we added a supplementary figure (now Figure 1—figure supplement 1) in order to summarize all the analyses (qualitative and quantitative) performed in our report and the materials on which they were carried out.

Considering our aim (to know the relevance of cINs in the cortex and to establish their relative amount in all species studied, in a comparable way), we took advantage from the fact that cINs occur with a monolayer-like arrangement (within the layer II); hence, linear density cell count (DCX+ cells/mm of cortical layer II) was the best option to obtain comparable estimations in brains which highly differ in size (due to species or ages). Such density, calculated on the real cortical layer II length measured in entire brain coronal sections (highly varying in different species and ages), represents a comparable value, allowing inferences across different mammals. Moreover, counting “all” DCX+ cells in the entire layer II perimeter in brain coronal sections allowed us to recognize the two morphological types (type 1 and type 2 cells; see Figure 2D) and thus to obtain their relative percentage (see pie charts in Figure 2E).

As to the amount of cINs, we provided both the “relative” amount in each species (linear density) and an estimation of the “total” amount (the latter, of course increasing with increase of the brain size). The linear density was obtained after counting DCX+ cells on the entire layer II perimeter of 12 coronal sections (see Figure 1—figure supplement 1 and Figure 3—figure supplement 1 ) in each of the 80 brains analysed. This cannot be considered semi-quantitative.

The striking result is that by comparing small-brained and large-brained species a huge difference (one order of magnitude) is present in density, thus going well beyond any influence coming from different lifestyle or fixation procedure (we know that a slightly different shrinkage can occur after brain fixation, but, again, this is far from the difference we found between species; see also other points from other reviewers on this aspect). The fact that animal species with large brains have also the highest linear densities even further increases the total number of cINs available in those brains (the so-called “reservoir” – see also answer to Point 3). Figure 5A right, shows that the neocortex of a mouse can contain about one cINs on average, against the two million of a chimpanzee (data shown in Supplementary file 4).

As a final consideration, we would like to underline that is very rare to find studies addressing many species in a comparable way (trying to reduce to a minimum the unavoidable differences in the procedure – see discussion of point 1 from reviewer #1 -, most studies found in literature being performed on a single animal species, using methods which often highly differ between different Authors). Here we addressed the issue of the occurrence, distribution and amount of cINs with a thorough analysis, and found a solid, statistically significant difference.

Changes: We tried to better explain how and why the quantifications were done. A new supplementary figure was provided which summarizes all the analyses carried out in our work (Figure 1—figure supplement 1) and some sentences summarizing the above mentioned concepts were added in the Results, in the paragraph of quantifications.

Furthermore, early in the text the concept of linear density must be explained to avoid misunderstandings.

Changes: We agree that the concept of linear density must be explained earlier (now it is introduced at the end of the Introduction). We explained better why we choose linear density at the beginning of Results.

2) The coverage of the literature is rather selective and quite biased to a limited set of authors. Other studies about layer II neurogenesis (disputed or not) are not discussed (e.g. Bifari et al., 2017). The entire discussion appears to be deliberately somewhat detached from the rest of the field. This is rather strange given the fact that the work by Luzzati, Amrein and others makes important contributions to the key discussions and is widely appreciated. The authors should integrate their work with the emerging ideas in the entire field, including its controversies.

We think that the reviewer here is referring to papers mainly dealing with neurogenesis. We agree that references should be expanded (we did it, also in relation to other points raised by other reviewers, but we would like to maintain a clear focus on the cortical immature (non-newly generated) neurons, so as not to dilute communication of the findings that follow directly from our new data. We took suggestion from the point 4 (see below) by focusing more on cINs and cerebral cortex, and by reducing parallelisms with “classic” adult neurogenesis occurring in the neurogenic niches.

Nevertheless, besides current controversies we do not want to enter here, a difficulty (but also an opportunity) resides in the emerging fact/hypothesis that cINs are something in the middle between newly generated neurons (from stem cells in the neurogenic niches) and plastic cells which can possibly add to (integrate into) the circuits, notwithstanding they cannot proliferate during adulthood. So, they are linked to neurogenesis, but of a different type and in a different region (the cerebral cortex).

Changes: We tried to insert the above mentioned concept in the Discussion. We added Luzzati et al., 2003 and Luzzati, 2015. The image in Figure 5E was deleted.

(Note: Irmgard Amrein has always been working on adult neurogenesis in the hippocampus).

(Note: Bifari et al., 2017 describes cells reaching the cortex in the postnatal period and mainly located in lower layers (e.g., layer III-IV). Only some of these cells express low levels of DCX. In constructive discussions we had with Francesco Bifari (we wrote a project together) we agree that they should belong to different cell populations).

3) Issue #1 notwithstanding, the quantitative conclusions are not fully clear. With the extension of layer 2, the number of the immature neurons appears to increase. What would that mean?

This is indeed the striking result of this work: by increasing brain size, with consequent increase of layer II perimeter length (that is: generally moving from the small-brained to the large-brained species), the occurrence (density) of the immature neurons increases (one order of magnitude from mouse to large-brained species), thus showing they really have a “reservoir” and supporting our hypothesis that they are an additional alternative mode of plasticity for large-brained mammals.

Changes: We better explained how the neuron counts were performed and we added the Figure 1—figure supplement 1, as discussed in point 1.